# The impact of repeated rapid test strategies on the effectiveness of at-home antiviral treatments for SARS-CoV-2

Tigist F. Menkir[1,2] & Christl A. Donnelly [2,3] ✉

Regular rapid testing can provide twofold benefits: identifying infectious individuals and providing positive tests sufficiently early during infection that treatment with antivirals can effectively inhibit development of severe disease. Here, we provide a quantitative illustration of the extent of nirmatrelvir-associated treatment benefits that are accrued among high-risk populations when rapid tests are administered at various intervals. Strategies for which tests are administered more frequently are associated with greater reductions in the risk of hospitalization, with weighted risk ratios for testing every other day to once every 2 weeks ranging from 0.17 (95% CI: 0.11–0.28) to 0.77 (95% CI: 0.69–0.83) and correspondingly, higher proportions of the infected population benefiting from treatment, ranging from 0.26 (95% CI: 0.18–0.34) to 0.92 (95% CI: 0.80–0.98), respectively. Importantly, reduced treatment delays, coupled with increased test and treatment coverage, have a critical influence on average treatment benefits, confirming the significance of access.

Rapid tests for SARS-CoV-2 have been shown to help identify individuals who may be infectious[1–3]. Their newfound use, particularly among those prone to severe disease, is identifying infections when they can be most effectively treated with antiviral treatments, including the Pfizer drug PF-07321332 (nirmatrelvir)[4,5], which necessitates early use to lower the risk of hospitalization. Here, we demonstrate that testing rates, as well as testing and treatment coverage and positive-test-to-treatment delays, shape the impacts of such test-and-treat policies.

Many have promoted rapid testing to identify infections when antivirals are still helpful[6–8], so there is a need to quantify the extent to which frequent rapid testing can enable high-risk patients to benefit most from the treatment. Thus, we build on prior studies which have served to characterize the ability of rapid testing strategies to identify presymptomatic patients or to reduce transmission[9–13]. Specifically, acknowledging the short window over which treatment can effectively inhibit more severe outcomes, we assess different strategies—defined by varying rates of test administration—in their relative ability to curtail the risk of hospitalization in an adult patient population facing an increased risk of severe disease, i.e., those who would be offered treatment in the event of testing positive.

To evaluate the benefits of repeated rapid testing at different rates on treatment effects, we used inferred lateral flow test (LFT)-associated positivity estimates from a Hellewell et al. analysis[9] and estimated hospitalization risks when treated within 3 and 5 days following the onset of symptoms from the December summary of the Phase 2/3 EPIC-HR trial findings[4]. Specifically, for each rapid testing strategy (every other day, every 3 days, once a week, once every 2 weeks, strategies explored in ref. [10], and once only after symptom onset) we estimated test-positivity-probability-weighted risk ratios (RRs) of hospitalization—hereafter referred to as 'weighted RRs of hospitalization'—as a function of time since infection, the proportion of the infected population who would be offered the treatment, and the proportion of the infected population who would take it sufficiently early to benefit from treatment. In sum, to generate weighted RRs for each testing regime, we assigned probabilities for every possible testing sequence consistent with the regime, leveraging the Hellewell et al. positivity estimates[9] as a function of time since infection, to

[1]Center for Communicable Disease Dynamics, Department of Epidemiology, Harvard T.H. Chan School of Public Health, Harvard University, Boston, MA, USA. [2]Department of Statistics, Oxford University, Oxford, UK. [3]MRC Centre for Global Infectious Disease Analysis, Department of Infectious Disease Epidemiology, Imperial College London, London, UK. ✉e-mail: christl.donnelly@stats.ox.ac.uk

period-specific ratios comparing the risk of hospitalization in treatment and placebo groups, leveraging the EPIC-HR summary data.

To estimate the proportion of the infected population offered treatment under each testing regime, we again used the Hellewell et al. positivity estimates to yield the complement of the proportion of the population never testing positive over all possible testing sequences. Additionally, we estimated the proportion of the population who would be given treatment at a time when it is associated with a non-zero reduction in the risk of hospitalization in the same way we generated weighted RRs, instead weighting indicators of whether the test is conducted during the clinically relevant window. We further evaluated the proportion of the infected population offered/benefiting from treatment under a one-time testing strategy immediately following symptom onset. Finally, we explored the sensitivity of our findings to assumed treatment efficacy trends, an incubation period distribution more consistent with Omicron infections[14], and three measures of access: treatment uptake or coverage, the delay from testing positive to treatment, and testing coverage.

## Results

As expected, we found that when tests are administered more frequently, the benefits associated with nirmatrelvir initiation increase dramatically, such that treatment substantially reduces the risk of hospitalization (Fig. 1a). While the median RR associated with the every other day strategy is 0.17 (95% CI: 0.11–0.28), the median RR associated with the once every 2 weeks strategy is 0.77 (95% CI: 0.69–0.83), with a dramatic increase in median RRs from the two higher-frequency testing regimes to the less-frequent testing alternatives (Fig. 1a). Correspondingly, we see a pronounced increase in the proportion of the

infected population benefiting from treatment as testing frequency increases, ranging from 0.26 (95% CI: 0.19–0.34) to 0.92 (95% CI: 0.80–0.98) (Fig. 1b). The estimates of proportion given the treatment and proportion actually deriving some benefit from it indicate that nearly everyone who tests positive and thus takes treatment receives some benefit. This arises because, in the estimated Hellewell et al. positivity curves[9], nearly all positive tests occur within 2 weeks of infection. Consequently, under our base case scenario, where drug-associated benefits extend to 7 days since symptom onset (which corresponds to 12 days since infection assuming an incubation period of 5 days), almost all individuals who test positive are captured within this drug efficacy window.

Under a sensitivity analysis in which we assume a distribution of shorter incubation times, to reflect time-to-symptom-onset trends for patients infected with the Omicron variant, our estimates indicate that fewer individuals are able to benefit from treatment (Supplementary Table S1), as expected when symptoms develop more quickly and there is a reduced opportunity to test when treatment is more effective. However, we note that any differences in estimated proportions benefited between the two scenarios are modest (Supplementary Table S1).

In comparison to the multi-frequency testing strategies, an approach of testing once after symptoms arise results in a notable proportion of the infected population given treatment, but with substantial variability (0.51, 95% CI: (0.30–0.80) and 0.42, 95% CI: (0.080–0.80) for our baseline scenario and shorter incubation period scenario, respectively) (Fig. 1 and S2). Importantly, while this strategy - for the baseline scenario - was found to outcompete the lower-frequency strategies of testing every week and every 2 weeks, its

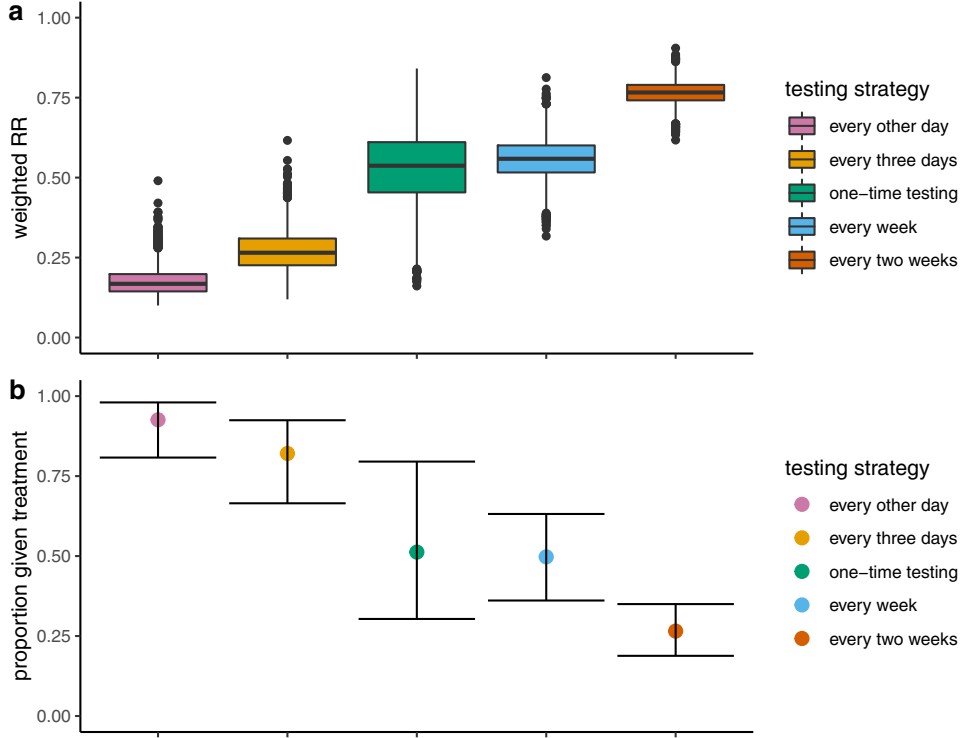

**Fig. 1 | Estimated weighted hospitalization risk ratios (weighted RR), relative to the patient population untreated with nirmatrelvir and estimated proportions treated, by testing strategy. The weighting of RRs reflects the likelihood of testing positive and therefore being treated with nirmatrelvir x days after becoming infected. a** Distribution of estimated weighted RRs of hospitalization by testing strategy: every other day (pink), every 3 days (orange), one-time testing (green), every week (blue), and every 2 weeks (dark orange). Medians are marked by solid horizontal lines, each box includes the full interquartile range, and plotted points are those which extend beyond the upper/lower quartile +/− 1.5*interquartile range. n = 4000 MCMC samples, each consisting of positivity estimates up to 30 days since infection[9]. **b** Estimated median proportions given treatment by testing strategy (including the one-time post-symptom onset testing strategy) with 95% CIs. In all cases no positive-test-to-treatment delay and full test coverage were assumed. As before, n = 4000 MCMC samples, each consisting of positivity estimates up to 30 days since infection[9].

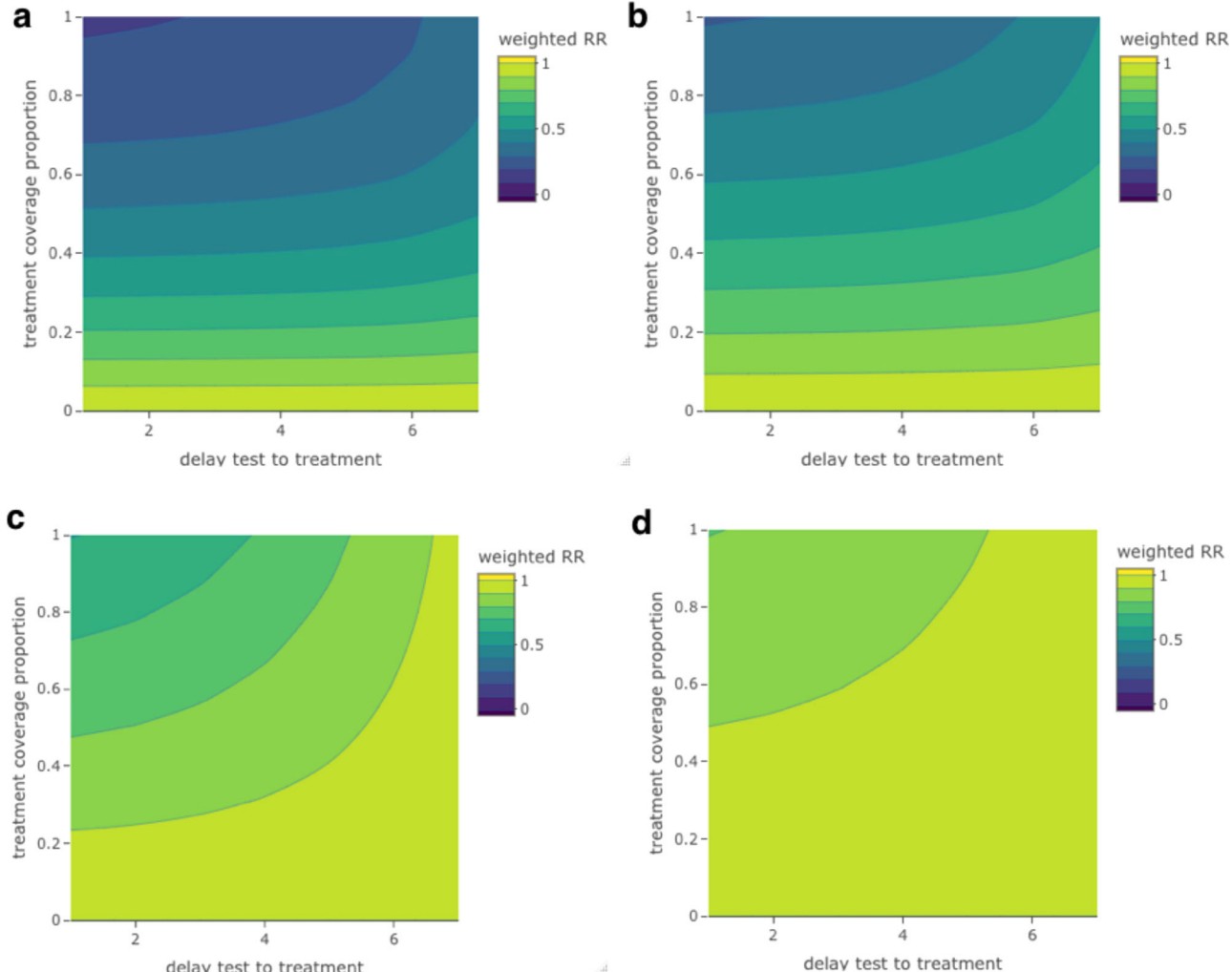

**Fig. 2 | Sensitivity of estimated weighted hospitalization risk ratios (weighted RR), relative to the patient population untreated with nirmatrelvir, to treatment coverage and treatment delays, across testing strategies. The weighting of RRs reflects the likelihood of testing positive and therefore being treated with nirmatrelvir x days after becoming infected.** Sensitivity of estimated weighted RRs of hospitalization to positive-test-to-treatment delays (*x*-axis) up to 7 days and treatment coverage proportions (*y*-axis) up to full coverage, by testing strategy (**a** every other day, **b** every 3 days, **c** every week, **d** every 2 weeks). Darker colors indicate lower weighted RRs, i.e., greater treatment-associated reductions in hospitalization in risks.

expected impact is far eclipsed by the every-other-day and every-3-days strategies, with only the latter two enabling a strong majority of the population to be offered treatment. We note that under the shorter incubation scenario, however, the one-time testing strategy instead reports a lower proportion offered treatment than the once-every-week strategy. Our baseline results highlight the essential trade-offs between testing costs and treatment impacts; despite the increased investment that would be required for more-frequent testing, a vastly increased proportion of the at-risk population would be afforded the opportunity to benefit from treatment. Furthermore, that the weekly and bi-weekly testing regimes are generally less effective than simply testing once symptoms emerge highlights that more-frequent testing is essential for any repeated testing policy to have any real added treatment-associated benefits. These findings replicate what have been observed in prior studies about the crucial role of "test frequency" in the transmission-limiting context[10,11,13], suggesting that such regular testing regimes would assure benefits in both infection prevention and disease control. A potential hybrid testing scheme might also be worth considering, with less frequent repeated testing as well as a test immediately following the onset of symptoms, should they occur, which would provide some intermediate benefit at some intermediate cost than its more or less frequent testing counterparts.

We found that treatment benefits depend on both treatment and test coverage and the delay from testing positive to treatment (Fig. 2). To achieve RRs within the range of what we observed with full coverage, zero delays and testing every other day, treatment coverage of at least 70% would require positive-test-to-treatment delays of no more than 2 days. With more sparse testing, treatment coverage and positive-test-to-treatment delays are critical, with smaller RRs achieved only through nearly full coverage and delays of no more than 2 days. When we independently assess the impacts of testing coverage, we find that estimated proportions benefiting from treatment are particularly sensitive to the assumed proportion testing, particularly for the more-frequent-testing strategies (Supplementary Fig. S3). However, we find that when we assume a high test coverage, the less-frequent-testing strategies are broadly outperformed by their more-frequent counterparts under low test coverage (Supplementary Fig. S3).

Based on the hospitalization risks at the two treatment initiation time ranges considered in the Phase 2/3 EPIC-HR trial, we fitted RRs and assumed a linear decline in efficacy to estimate the treatment efficacy levels associated with nirmatrelvir treatment across a range of days since symptom onset. To vary these assumptions, we considered trends that could capture two different time windows of efficacy

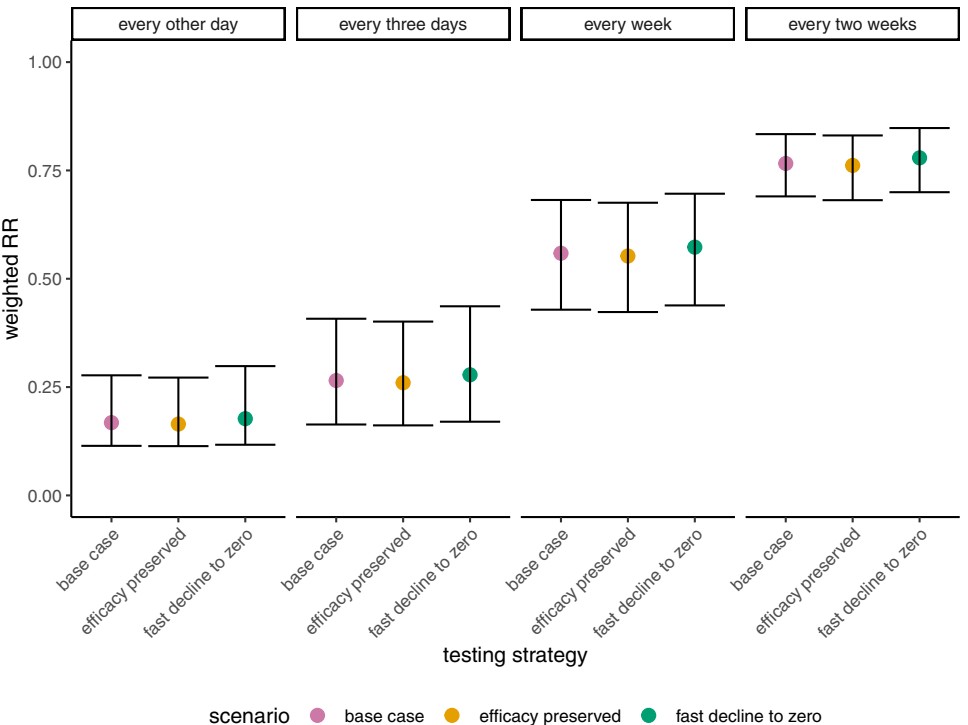

**Fig. 3 | Sensitivity of estimated weighted hospitalization risk ratios (weighted RR), relative to the patient population untreated with nirmatrelvir, to treatment coverage and treatment delays to assumed treatment efficacy trends, across testing strategies. The weighting of RRs reflects the likelihood of testing positive and therefore being treated with nirmatrelvir x days after becoming infected.** Median estimated weighted RRs of hospitalization by assumed treatment efficacy scenario (base case scenario (pink), scenario with preserved efficacy following 5 days after symptom onset (orange), and scenario with efficacy dropping to 0 following days after symptomonset (green)) across testing strategies: every other day, every 3 days, every week, and every 2 weeks. In all cases no positive-test-totreatment delay and full test coverage were assumed. We do not assess the additional efficacy scenarios for the one-time testing strategy, because under this strategy, individuals who test positive take treatment on the day of testing; as such, the assumed RR trends beyond 0 days since symptom onset are irrelevant. Error bars indicate the 95% CI around each estimate. n = 4000 MCMC samples, each consisting of positivity estimates up to 30 days since infection[9].

beyond the range considered in the trial, and found little to no changes in our estimated RRs (Fig. 3). We further note that while non-linear trends may marginally alter the magnitude of our expected RRs, with RRs inflated towards 1 if we assume a curvilinear decline consistent with a shorter efficacy window, they are unlikely to change the observed relative magnitude across strategies.

The positivity data from the Hellewell et al. analysis assumed an "LFT-like" cycle threshold (CT) of 28[9]. If a lower CT threshold were used, we would expect the estimated RRs of hospitalization to increase and the corresponding proportion benefiting from the treatment to decrease, with the converse holding true for a higher assumed CT threshold. However, the ordering of RRs across strategies would once again persist. Patient data were collected in early 2020, such that time-specific positivity estimates were obtained from wild-type infections, with trends that could differ from the currently predominant variant[9]. In contrast, hospitalization risks were estimated using data from July 2021 and thus likely were recorded on largely Delta-infected patients[5]. We note that if the prevailing variant were associated with a substantially increased or reduced risk of hospitalization, this would likely hold true for patients in general, regardless of whether they received treatment, such that the relative risks of hospitalization would remain relatively unchanged. If, however, treatment is effective for a longer (or shorter) period of time, we would observe a narrowing (or expanding) benefit of more periodic testing. Additionally, if treatment were found to be effective under the same time frame, but to a greater extent, we would anticipate increased benefits under all testing strategies. Thus, it is important to update our results specific to the current variant and among vaccinated populations[5], once new data become available.

## Discussion

Despite the promising role of rapid testing that we observe here, it is important to acknowledge the costs that result from such testing policies. For instance, approximately six billion pounds have been paid by the UK government for their mass lateral flow distribution plan, which concluded on April 1, 2022[15]. However, we note that under a focused testing plan, prioritizing frequent testing among those who are most likely to be prescribed treatment upon a positive test, as is the subject of attention here, these costs would be considerably less. From the perspective of patient populations[16], costs include those associated with (highly unlikely[17]) false positive results, such as missed earnings from work, missed medical appointments for other health conditions, and stress-related mental health consequences[18]. However, with a substantial proportion of the population successfully being linked to treatment due to testing, there may be significant cost savings (to both hospitals and patients) from averted hospitalizations, specifically among patients who may be driven to debt as a result of these expenses[19–21].

In sum, we characterized how rapid testing may facilitate treatment benefits among those most likely to be hospitalized, with more frequent testing yielding the best results. While we also observed notable benefits under a one-time test policy, this regime requires that individuals recognize symptoms and, as with the other strategies, have tests available to use soon after symptoms emerge. Test and treatment access matters: high coverage and short delays from testing to treatment are necessary to achieve large benefits. Spatially-refined testing strategies might further support disadvantaged communities where vulnerabilities to severe disease and barriers to testing and treatment are most concentrated. Finally, regular testing is potentially cost-

saving, particularly in high-prevalence settings, as it is associated with dramatically reducing hospitalizations, which may outweigh the costs of testing and treatment distribution.

## Methods

Our analysis required two primary sources of data to estimate probability-weighted risk ratios (RRs): positivity curves and treatment efficacy-associated RRs. To parameterize the test positive probabilities, we used an estimated lateral flow test "(LFT)-like" positivity curve generated from a Hellewell et al. Bayesian model, which reports the posterior probability of testing positive for each day since infection from 0 to 30 days, across 4000 Markov chain Monte Carlo (MCMC) draws[9]. That is, while the patient population in this analysis were routinely administered PCR - and not antigen - tests, the authors provided two scenarios estimating expected test positive probabilities under antigen testing, by defining lower CT thresholds that would better reflect the diagnostic ability of LFTs. To parameterize RRs of hospitalization, we used data reported in the latest EPIC-HR press release, namely the number of hospitalized subjects in the treatment and placebo groups, and the total size of each treatment arm, stratified by when treatment was initiated, i.e., within 3 or 5 days since symptom onset, in a trial population of SARS-CoV-2-positive patients at an elevated risk of severe disease[4,5].

### Estimating weighted RRs, proportion given and benefiting from the treatment

To estimate RRs over a range of days since symptom onset, we fit a log-binomial model to the reported hospitalization counts by assigning a series of treatment-labeled 1s and 0s to reflect the recorded hospitalizations and non-hospitalizations within each arm, with covariates representing days since symptom onset and treatment group. We assumed a linear relationship between the continuous predictors (day and the treatment x day interaction) and the log risk of hospitalization as well as an additive effect of treatment. From our model fit, we extracted RR estimates for each day on the range [0,7] days since symptom onset and defined a RR of 1 (i.e., no treatment effect) thereafter, given the limited, if any, efficacy that is expected to occur a week following symptom onset. We additionally fit a logistic model to determine whether our results were sensitive to whether RRs and ORs were employed, and observed nearly equivalent results, as expected given the rarity of the hospitalization outcome (Fig. S1).

Consistent with the method detailed in Hellewell et al. for estimating probabilities of pre-symptom onset symptomatic detection[9], assuming a CT threshold of 28, we used the probability of either first testing positive at the testing day $t_{max}$ (the product of the test negative probabilities for each test day up until day $t_{max}$ and the test positive probability at day $t_{max}$) to weight, among the treated, the corresponding RR at day $t_{max}$, or testing negative at all testing days, weighted by an RR of 1 corresponding to no treatment; we then averaged these weighted RRs over all possible testing sequences, as shown in Eq. (1). We replicated this process over the 4000 simulations of positivity data and for each testing strategy (every other day, every 3 days, once a week, and once every 2 weeks). To summarize our results for each strategy, we extracted median weighted RRs and an accompanying 95% confidence interval (CI). To match hospitalization risks reported with the time reference of days since symptom onset to test positive probabilities reported with the time reference of days since infection, we sampled incubation periods (the times from infection to symptom onset) - assuming a lognormal distribution consistent with pooled estimates from a McAloon et al. meta-analysis[22] - at each iteration and used the rounded incubation periods to index time relative to symptom onset rather than infection and identify the appropriate RR for each day. We allowed for possible delays in x days from testing positive to treatment by additionally setting a variable which shifts this period by x days. In our main analyses, we assumed a zero-day positive-test-to-treatment delay corresponding to potential plans to distribute antiviral pills to households to store for immediate use if necessary and further assumed full treatment coverage.

$$Weighted\ RR_{strategy\,ts} = \frac{1}{index}\left\{ \sum_{seq\,\epsilon\,tseq} \prod_t^{t_{max}-1}[\Pr(test-)_t * \Pr(test+)_{t_{max}} \right.$$
$$\left. * RR_{t_{max}+delay-incubation.period} * P_{tmnt}] + \sum_{seq2\,\epsilon\,tseq2}\left[\prod_{t2}^{t2_{max}}\Pr(test-)_{t2}*1\right] \right\} \quad (1)$$

where

$seq\,\epsilon\,tseq$ denotes a given testing sequence seq - each defined by when testing is initiated - in the full set of possible sequences consistent with strategy ts, t is initialized as the first time point for each testing sequence, $t_{max}$ is the final day of that testing sequence, $RR_{t_{max}+delay-incubation.period}$ denotes the risk ratio associated with the final day of that testing sequence, back-shifted for the sampled incubation period, adding any test-to-treatment delays, $P_{tmnt}$ denotes the the treatment proportion, $seq2\,\epsilon\,tseq2$ denotes a given testing sequence in the set of always-test-negative sequences consistent with strategy ts t2 is initialized as the first time point for each always-test-negative testing sequence, and $t2_{max}$ is the final time point of that testing sequence, and index = 2, 3, 7, and 14 for every other day, every 3 days, once a week, and once every 2 weeks, respectively

We subsequently estimated the proportion of the infected population who could be offered the treatment (which can be interpreted as "given the treatment" under full treatment coverage and uptake), representing all infected individuals who test positive at any point. Specifically, this proportion can be expressed as the complement of the mean across testing sequences of the probability of testing negative at all time points, which captures all always-test-negative possibilities, standardized by the number of possible sequences, as described in Eq. (2) below.

$$Proportion\ offered\ treatment_{strategy\,ts} = 1 - \frac{1}{index}\sum_{seq\,\epsilon\,tseq}\prod_t^{t_{max}}\Pr(test-)_t \quad (2)$$

Finally, we estimated the proportion of the infected population who derive some benefit from treatment, that is, those able to initiate treatment during a period when the RR does not equal 1. To do so, for each possible testing sequence, we first assigned an indicator variable equal to 1 for the positive testing time if it precedes the day when RR = 1 and equal to 0 otherwise (which incorporates those never testing positive and those testing positive too late to benefit from treatment). We then averaged test-positive-probability-weighted indicators across all testing sequences to give the mean proportion benefiting from the treatment. $I(day_i < = 7 + incubation.period)$ denotes whether the testing day plus any test-to-treatment delay falls within 7 days of the incubation period (that is, 7 days following symptom onset), when we assume treatment is no longer efficacious. To account for imperfect testing, we multiplicatively reduced our estimated proportions benefiting from treatment by the proportion testing.

$$Proportion\ benefiting_{strategy\,ts} = P_{test} * \frac{1}{index}\sum_{seq\,\epsilon\,tseq}\left\{\prod_t^{t_{max}-1}\Pr(test-)_t \right.$$
$$\left. *\Pr(test+)_{t_{max}} * [I(day_i<=7+incubation.period)] \right\} \quad (3)$$

where $P_{test}$ denotes the testing proportion.

In the context of a strategy of testing once only after once symptoms develop, assuming 0-day positive-test-to-treatment delays, individuals will always take treatment on the day of their (positive) test result and thus, (1) all those who are offered treatment benefit from it and (2) there is only one possible testing probability, with a corresponding risk ratio of hospitalization that is always that of 0 days since

symptom onset. As before, we report medians and a corresponding 95% CI to summarize estimates across the 4000 samples.

### Sensitivity analyses

To accommodate shorter incubation periods that have been observed among Omicron infections, we considered a Weibull incubation period distribution found to generate the optimal fit to symptom onset times in an analysis of Omicron-infected subjects in Norway[14]. Our outcomes of focus in this sensitivity analysis are weighted RRs and proportions benefited under each testing strategy (Supplementary Fig. S2), as the estimated proportions offered treatment, which do not depend on when an individual tests positive, will remain unchanged.

We additionally evaluated how varying our assumed period of treatment efficacy, reflecting waning of treatment impacts, would influence the marginal benefits of more frequent rapid testing. In addition to our base case scenario, where we assumed a linear decline in efficacy, as described previously, we considered two additional scenarios assuming either (1) the RR consistent with initiation at 5 days since symptom onset is preserved following 5 days after symptom onset until the latest positive test (referred to as "efficacy preserved" in Fig. 3) or (2) the RR goes to one, reflecting zero efficacy, after 5 days since symptom onset (referred to as "fast decline to zero" in Fig. 2). Weighted RRs are estimated for each scenario (Fig. 3).

We subsequently considered the combined impacts of treatment coverage and positive-test-to-treatment delays on weighted RRs by jointly varying the treatment coverage proportion across the range [0,1] by 0.1 (10%) unit increments and the delay from testing positive to treatment across the range [0, 7] days, under our main treatment efficacy scenario. For this, we used median estimated day-specific test positivity values. We then compared weighted RRs across all resulting combinations (Fig. 2). As described previously, in a separate analysis, we examined the isolated impacts of test access on proportions offered and benefiting from treatment through a series of scenarios in which we varied test coverage across a range of values from 0 to 1 (Supplementary Fig. S3).

### Reporting summary

Further information on research design is available in the Nature Research Reporting Summary linked to this article.

## Data availability

LFT positivity data were generated by implementing the relevant scripts in open source code provided in https://github.com/cmmid/pcr-profile, based on the following paper by Hellewell et al.: https://bmcmedicine.biomedcentral.com/track/pdf/10.1186/s12916-021-01982-x.pdf. Pfizer drug efficacy data were obtained from the latest EPIC-HR summary release at https://www.pfizer.com/news/press-release/press-release-detail/pfizer-announces-additional-phase-23-study-results. The output generated in this study is publically available here: https://github.com/goshgondar2018/LFT_treatment_analysis/tree/main/output.

## Code availability

All code used to conduct are analyses is publically available here: https://github.com/goshgondar2018/LFT_treatment_analysis.

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

## Acknowledgements

T.F.M. acknowledges support from the Harvard Center for Communicable Disease Dynamics' Program Support fund and the Harvard T.H. Chan School of Public Health's Rose Traveling Fellowship. C.A.D. acknowledges support from the MRC Center for Global Infectious Disease Analysis, the NIHR Health Protection Research Unit in Emerging and Zoonotic Infections and the NIHR-funded Vaccine Efficacy Evaluation for Priority Emerging Diseases (PR-OD-1017-20007). We thank Fabian Falck, Christopher Williams, and Tim Reichelt for helpful feedback on a figure. The computations in this paper were run on the FASRC Cannon cluster supported by the FAS Division of Science Research Computing Group at Harvard University.

## Author contributions

T.F.M. and C.A.D. designed the study and C.A.D. provided supervision. T.F.M. performed the analyses and drafted the manuscript and C.A.D. reviewed and approved all drafts.

## Competing interests

The authors declare no competing interests.
