## [Peer Review File · Nature Communications]

The impact of repeated rapid test strategies on the effectiveness of at-home antiviral treatments for SARS-CoV-2REVIEWER COMMENTS

Reviewer #1 (Remarks to the Author):

A mathematical model was used to quantify the benefit of serial testing in administering antiviral treatment of SARS CoV-2 infection in reducing hospitalization. With testing every two days, the relative risk (relative to no treatment) was estimated to be 0.17 and 0.77 with testing every 14 days. Many of my comments revolve around improving the readability/clarity of the methodology and presentation.

1.The equations in the Supplement do not follow standard notation for summation and product indexing. This issue made the equations difficult to read and assess properly and may/may not require additional revisions after addressing this issue. Many of my following comments may stem from this issue.

2.Equation 1. The initial term " $[Pr(test +)]_i * R Ri$ " is not multiplied by the treatment proportion when the individual has a positive test.

3.Equation 1. It is unclear if the index variable i follows the sequence $\{0,1,2,\dots, index\}$ or $\{0,1,2,\dots, index-1\}$. If the sequence is $\{0,1,2,\dots, index\}$ then it would be incorrect because of the inclusion of an additional point; as the sequence would include "index+1" time points, instead of "index" time points.

4.In the Supplement, there is mention that the model can include an x -day delay from test to treatment into the calculations. However, none of the supplementary equations contain this parameter.

5.Equation 2. It is unclear why the use of the average in calculating the probability of testing negative at all time points. I expect the average accounts for the uncertainty in the time of infection between tests.

6.Equation 3. The supplementary text should explicitly define the indicator function $I(\cdot)$.

7.I would suggest the removal of Supplemental Figure 4, as it is repetitive of Supplemental Figure 3.

8.Figure 3 and Figure S3. Currently, each panel has specific colour axes limits (i.e., the z-axis differs for each sub-panel). I would suggest each sub-panel have the same z-scale for the effects across all scenarios can be better and more easily compared.

9.The captions for Figure S2 and S3 should have additional details to make them distinct from Figure 2 and Figure 3 in the main text. Currently, the captions describing Figure 2 and Figure S2 are the same. Similarly, the captions for Figure 3 and Figure S3 are the same.

10.Adding the sensitivity analyses scenarios described in the Supplement to Figure S1 would improve clarity about the descriptions provided in the text.

11.Figure 2 and Supplementary Figure S2 appear to be the same. Similarly, there seem to be no differences between Figure 3 and Figure S3.

12.Figure 4 is referenced in the main text, but Figure 4 is not presented.

13.Figure 1. Aspects of the box plot or error bars are not defined in the figure caption

14.Figure 1. Panels A) and B) are not provided in the figure

15.The scenario examined in the manuscript focuses on a situation where serial testing is conducted in a population, and individuals receiving a positive test would receive treatment. Serial testing would likely not be feasible on a large scale. One could examine the additional scenario of a

one-time test once symptoms appear. This analysis will better highlight the benefit\downfall of the different serial testing strategies.

16. Delta should be capitalized for "delta-infected patients".

17. The authors discuss the benefits of frequent testing. However, they should also expand on the potential drawbacks of regular testing and the implications (e.g., false positives)
The analysis is based on an incubation period of five days. There should be discussion or additional analysis on the effects of a shorter incubation period. A shorter incubation period means that treatment will need to be administered sooner from the time of infection to have any benefit.

Reviewer #2 (Remarks to the Author):

The analysis by Menkir and Donnelly addresses an important question, appears carefully and appropriately conducted and is clearly presented.

The only significant change to the analysis that could be considered would be to try to make the analysis more relevant to the currently circulating variant (i.e. Omicron), although I do acknowledge this is difficult. If this were done, the input that would clearly need to be changed would be the incubation period estimates (currently taken from McAloon). Omicron has a markedly shorter incubation period, which has been estimated to be less than half of wild-type and this would significantly change the estimates reported. While adapting the analysis for Omicron would improve the paper's applicability to the current epidemic, I fully acknowledge that this is a slippery slope. For example, the paper might still be criticised for not adapting all parameters to Omicron if the incubation period is changed, new variants might emerge while the revisions were being made, etc. Overall, I would lean towards including a sensitivity analysis with the incubation period adapted to Omicron, but am happy to be guided by the authors, other reviewers and editors. The data used by Hellewell et al. to estimate probability of test positivity by time from infection is from SAFER participants who eventually developed symptoms. Therefore, the entire analysis should consider only symptomatic persons in the pre-symptomatic phase. I believe this indeed exactly what the analysis does do, which is completely appropriate given that the outcome of interest is hospitalisation (which would only occur in symptomatic persons). However, in the second paragraph of the main text, it is stated "to identify presymptomatic and/or asymptomatic patients" and similarly in the second paragraph of Section A. of the Supplement it is stated "or estimating probabilities of asymptomatic and pre-symptom onset symptomatic detection". This wording seems misleading. If my understanding is correct and is essentially looking for pre-symptomatic infectiousness, then I have no issue with the approach, but I would suggest removing the reference to "asymptomatic" patients in the text.

I have spent some time looking at Equation 1, and I think I just about understand it, and I also believe it is correct. However, it is quite hard to fully understand from the equation and it places a large burden on the reader (as do the other equations to a lesser extent). While there is some explanation in the text, I believe it would be improved if the intuition of the terms could be explained a little further (in the Supplement). On a minor point, the square brackets in these equations appear redundant.

Similarly, I find the term "estimated test-positivity-probability-weighted risk ratios" hard to understand immediately, and it does not convey that these are essentially risk ratios for hospitalisation. I would think the paper would be clearer if the full term is only used in full at first usage, and abbreviated to something like "hospitalisation RRs" for subsequent instances.

ABSTRACT

Given the authors have performed such a detailed quantitative analysis, I would suggest including some specific numbers for the main findings here.

TEXT

I understood Paxlovid now has a generic name, nirmatrelvir, in which case this should be preferred.

Third paragraph – I would prefer "Hellewell estimates" (or equivalent) over "Hellewell data" throughout, given that I believe the authors are using the modelled profiles of test positivity probability over time.

Fourth paragraph – "The estimates of proportion", missing the word "the".

FIGURE 1

I note that the colours are not essential for this figure because the same message could be conveyed through labelling of the x-axis. (I don't feel strongly about this minor editorial issue).

FIGURE 2

The scenarios referred to here are explained slightly more in the sixth paragraph and this figure seems more relevant to that text (whereas this figure is referenced in the fourth paragraph). Suggest reordering to avoid this.

FIGURE 3

I would think this figure would be clearer and provide a fairer comparison if the colourbar was fixed across all panels.

SUPPLEMENTARY FIGURES

Supplementary Figures S2 and S3 are identical to Figures 2 and 3 (in the main text) and should be removed.

DISCLOSURE OF IDENTITY

I do not support single-blinded peer review and so choose to disclose my identity to the authors (because I am aware of theirs).

James Trauer, Monash University

Reviewer #3 (Remarks to the Author):

In the current manuscript, the authors evaluate the frequency of testing at which potential antiviral treatments such as Paxlovid can provide substantial population-level impact by reducing hospitalizations. Given that these treatment regimes have to be administered early in the infection for it to reduce the severity of the disease, the analysis presented has of important value. My comments on the paper are as follows:

1. Hellwell data used is based on RT-PCR testing. Please state it clearly in the manuscript. It was also not clear what turnaround time for results was considered for the analysis, as that will determine how quickly people begin treatment.
2. The best strategy identified by the authors is testing every other day. The proportion of infected people who use drugs during this best strategy is the highest. While at the individual level, it makes sense. At the population level it may not be feasible. Therefore, in terms of policy, it is important to discuss who is being tested. So, an analysis that actually accounts for the proportion of the population who may get tested (this could be the population that is more likely to get hospitalized for instance), will provide a better insight into the population-level utility of the treatment strategies.
3. In terms of feasibility, I am unsure why the author did not also evaluate the utilities of rapid antigen tests. I understand that Hellwell data was based on RT-PCR testing, however diagnostic sensitivity of rapid antigen tests relative to RT-PCR tests can be used to create a mapping to get relevant data for rapid antigen tests. The addition of such analysis will make findings from this study more valuable.
4. Were there any adjustments done to the Hellwell data to account for various or most current variants? Incubation periods, as well as the severity of the disease, vary by variant. Either accounting for them explicitly using estimates from literature or discussing the implications of not including them in the analysis is important.

Response to referees

Reviewer #1 (Remarks to the Author):

A mathematical model was used to quantify the benefit of serial testing in administering antiviral treatment of SARS CoV-2 infection in reducing hospitalization. With testing every two days, the relative risk (relative to no treatment) was estimated to be 0.17 and 0.77 with testing every 14 days. Many of my comments revolve around improving the readability/clarity of the methodology and presentation.

We thank the reviewer for the constructive feedback regarding the presentation of our methods and results.

1. The equations in the Supplement do not follow standard notation for summation and product indexing. This issue made the equations difficult to read and assess properly and may/may not require additional revisions after addressing this issue. Many of my following comments may stem from this issue.

We have updated our equations to follow a more standard and easy-to-follow format.

2. Equation 1. The initial term " $[Pr(test +)_i] * R Ri$ " is not multiplied by the treatment proportion when the individual has a positive test.

We now include the treatment proportion parameter in Equation 1.

3. Equation 1. It is unclear if the index variable i follows the sequence $\{0, 1, 2, \dots, \text{index}\}$ or $\{0, 1, 2, \dots, \text{index}-1\}$. If the sequence is $\{0, 1, 2, \dots, \text{index}\}$ then it would be incorrect because of the inclusion of an additional point; as the sequence would include "index+1" time points, instead of "index" time points.

We have removed the indexing we previously used in Equation 1, instead referencing each series of testing times in general terms as a possible testing sequence, rather than explicitly defining all the days included in each sequence.

4. In the Supplement, there is mention that the model can include an x -day delay from test to treatment into the calculations. However, none of the supplementary equations contain this parameter.

We now include the test-to-treatment delay parameter in equations 1 and 3 (the proportion offered treatment captures all subjects who test positive at any point and thus does not depend on test-to-treatment delays).

5. Equation 2. It is unclear why the use of the average in calculating the probability of testing negative at all time points. I expect the average accounts for the uncertainty in the time of infection between tests.

We have now clarified in our writing that we take the mean of testing negative at all time points to capture all possible testing sequences, i.e. testing negative at all time points defined in

sequence 1 or testing negative at all time points defined in sequence 2 etc., standardized by the number of possible testing sequences

6. Equation 3. The supplementary text should explicitly define the indicator function $I(\cdot)$.

We now define the indicator function in our introduction to equation 3.

7. I would suggest the removal of Supplemental Figure 4, as it is repetitive of Supplemental Figure 3.

We agree with this suggestion and have removed Supplemental Figure 4.

8. Figure 3 and Figure S3. Currently, each panel has specific colour axes limits (i.e., the z-axis differs for each sub-panel). I would suggest each sub-panel have the same z-scale for the effects across all scenarios can be better and more easily compared.

Thank you for highlighting this; we have standardized the color scale for all sub-panels.

9. The captions for Figure S2 and S3 should have additional details to make them distinct from Figure 2 and Figure 3 in the main text. Currently, the captions describing Figure 2 and Figure S2 are the same. Similarly, the captions for Figure 3 and Figure S3 are the same.

We have removed Figures S2 and S3 as they are identical to Figures 2 and 3 and now link any mentions of the two figures in the supplement to the main text figures.

10. Adding the sensitivity analyses scenarios described in the Supplement to Figure S1 would improve clarity about the descriptions provided in the text.

Thank you for this suggestion. We now include colored lines to indicate our two sensitivity analysis scenarios ('fast decline to zero' and 'efficacy preserved') in Supplementary Figures S1(A) and (C). We note that figures S3(B) and (D) remain unchanged as these two scenarios are defined by varying assumptions about the magnitude of relative risks over time, as opposed to absolute risks among the treated.

11. Figure 2 and Supplementary Figure S2 appear to be the same. Similarly, there seem to be no differences between Figure 3 and Figure S3.

These figures were indeed identical, as noted above; we had previously included them to be able to directly reference the figures in the supplement. However, to avoid any confusion, we now only include one version of the figures in the main text.

12. Figure 4 is referenced in the main text, but Figure 4 is not presented.

We removed the errant mention of Figure 4 in the main text.

13. Figure 1. Aspects of the box plot or error bars are not defined in the figure caption

We have expanded our legend for Figure 1 to describe the salient features of the box plot (i.e. the median, box, and whiskers) and error bars.

14. Figure 1. Panels A) and B) are not provided in the figure

We now include labels for Panels A and B in Figure 1.

15. The scenario examined in the manuscript focuses on a situation where serial testing is conducted in a population, and individuals receiving a positive test would receive treatment. Serial testing would likely not be feasible on a large scale. One could examine the additional scenario of a one-time test once symptoms appear. This analysis will better highlight the benefit/downfall of the different serial testing strategies.

We thank the reviewer for this valuable suggestion. To address this, we now include a scenario representing patients who test only one time following the onset of symptoms. To do so, we simply extract test positive possibilities at the conclusion of a sampled incubation period and summarize across iterations. We find that this strategy actually outperforms less-frequent strategies like testing every week and every two weeks in our baseline scenario, and note that this finding highlights how crucial shorter testing intervals are for any recurrent testing regime to be effective and consequently, for their costs to be justified.

16. Delta should be capitalized for “delta-infected patients”.

We have capitalized all references to Delta, such as this one.

17. The authors discuss the benefits of frequent testing. However, they should also expand on the potential drawbacks of regular testing and the implications (e.g., false positives)

We agree that facilitating a balanced conversation about the utility of frequent testing is important and now include a discussion on the potential downsides of such approaches in the penultimate paragraph of our main text. Specifically, we discuss the costs of widespread testing (with an example for the UK), noting that these costs may be attenuated under a focused testing plan and with an increasing share of hospitalizations averted. We additionally acknowledge the economic, physical and mental health-associated consequences of false positive test results, although note that such false positive outcomes are rare for most antigen tests.

18. The analysis is based on an incubation period of five days. There should be discussion or additional analysis on the effects of a shorter incubation period. A shorter incubation period means that treatment will need to be administered sooner from the time of infection to have any benefit.

To address this critical issue you raise, we have added a sensitivity analysis to reflect a shorter Omicron-like incubation period. We found that our estimated weighted RRs were indeed greater (and proportion benefiting from treatment lower) than those under our baseline scenario, given that as you note, with a shorter time to developing symptoms, a greater share of positive tests occur when treatment is less effective. However, we remark that the difference in estimates of weighted RRs and proportion benefited between the two scenarios is largely negligible.

Reviewer #2 (Remarks to the Author):

The analysis by Menkir and Donnelly addresses an important question, appears carefully and appropriately conducted and is clearly presented.

We thank the reviewer for this positive assessment of the focus, content, and presentation of our work.

The only significant change to the analysis that could be considered would be to try to make the analysis more relevant to the currently circulating variant (i.e. Omicron), although I do acknowledge this is difficult. If this were done, the input that would clearly need to be changed would be the incubation period estimates (currently taken from McAloon). Omicron has a markedly shorter incubation period, which has been estimated to be less than half of wild-type and this would significantly change the estimates reported. While adapting the analysis for Omicron would improve the paper's applicability to the current epidemic, I fully acknowledge that this is a slippery slope. For example, the paper might still be criticised for not adapting all parameters to Omicron if the incubation period is changed, new variants might emerge while the revisions were being made, etc. Overall, I would lean towards including a sensitivity analysis with the incubation period adapted to Omicron, but am happy to be guided by the authors, other reviewers and editors.

We appreciate this point about the relevance of our assumed incubation period and choose to include an additional sensitivity analysis assuming a shorter incubation period that would be more representative of Omicron infections. We find that while, as expected, estimates from the Omicron-associated scenario indicate that fewer participants can benefit from treatment, with a shorter time to symptom onset resulting in a lower proportion of positive tests occurring when treatment is more effective, any differences in estimated weighted risk ratios and proportion benefited are largely marginal.

The data used by Hellewell et al. to estimate probability of test positivity by time from infection is from SAFER participants who eventually developed symptoms. Therefore, the entire analysis should consider only symptomatic persons in the pre-symptomatic phase. I believe this indeed exactly what the analysis does do, which is completely appropriate given that the outcome of interest is hospitalisation (which would only occur in symptomatic persons). However, in the second paragraph of the main text, it is stated "to identify presymptomatic and/or asymptomatic patients" and similarly in the second paragraph of Section A. of the Supplement it is stated "or estimating probabilities of asymptomatic and pre-symptom onset symptomatic detection". This wording seems misleading. If my understanding is correct and is essentially looking for pre-symptomatic infectiousness, then I have no issue with the approach, but I would suggest removing the reference to "asymptomatic" patients in the text.

We thank the reviewer for identifying this critical nuance when specifying our population of interest. It is exactly the case that we are focusing on the subset of the population who will eventually develop symptoms and we adjust our language in both the second paragraph of the main text and third paragraph of the supplementary appendix to indicate this.

I have spent some time looking at Equation 1, and I think I just about understand it, and I also believe it is correct. However, it is quite hard to fully understand from the equation and it places a large burden on the reader (as do the other equations to a lesser extent). While there is some explanation in the text, I believe it would be improved if the intuition of the terms could be explained a little further (in the Supplement).

We have revised our Equation 1 to follow a more standard and easy-to-follow format in an effort to address any issues with clarity.

On a minor point, the square brackets in these equations appear redundant.

We have removed the square brackets in our updated version of Equation 1.

Similarly, I find the term “estimated test-positivity-probability-weighted risk ratios” hard to understand immediately, and it does not convey that these are essentially risk ratios for hospitalisation. I would think the paper would be clearer if the full term is only used in full at first usage, and abbreviated to something like “hospitalisation RRs” for subsequent instances.

We have amended how we reference this quantity in accordance with your suggestion, only using the phrase “test-positivity-probability-weighted risk ratios” when we first introduce it in the third paragraph of the main text and subsequently using the phrase “weighted risk ratios of hospitalization”.

ABSTRACT

Given the authors have performed such a detailed quantitative analysis, I would suggest including some specific numbers for the main findings here.

We agree that the inclusion of numeric results in our abstract is crucial and now include estimates of the range of estimates of the weighted risk ratios and proportion benefiting from treatment across strategies.

TEXT

I understood Paxlovid now has a generic name, nirmatrelvir, in which case this should be preferred.

We have replaced all mentions of Paxlovid with its generic name.

Third paragraph – I would prefer “Hellewell estimates” (or equivalent) over “Hellewell data” throughout, given that I believe the authors are using the modelled profiles of test positivity probability over time.

We agree with this point and now use the phrase “Hellewell estimates” in lieu of “Hellewell data”, as suggested.

Fourth paragraph – “The estimates of proportion”, missing the word “the”.

We have inserted the missing “the” here.

FIGURE 1

I note that the colours are not essential for this figure because the same message could be conveyed through labelling of the x-axis. (I don't feel strongly about this minor editorial issue).

We choose to preserve the strategy-specific coloring of the bars and points in Figure 2 for optimal ease of connecting identical strategies across panels.

FIGURE 2

The scenarios referred to here are explained slightly more in the sixth paragraph and this figure seems more relevant to that text (whereas this figure is referenced in the fourth paragraph). Suggest reordering to avoid this.

Following the reviewer's suggestion, we unite our discussion of the treatment efficacy scenarios and references to Figure 2, in the ninth paragraph of the main text.

FIGURE 3

I would think this figure would be clearer and provide a fairer comparison if the colourbar was fixed across all panels.

We thank the reviewer for this observation; we now provide a standard scale for the Z axis across panels in Figure 3.

SUPPLEMENTARY FIGURES

Supplementary Figures S2 and S3 are identical to Figures 2 and 3 (in the main text) and should be removed.

We have removed Supplementary Figures S2 and S3.

Reviewer #3 (Remarks to the Author):

In the current manuscript, the authors evaluate the frequency of testing at which potential antiviral treatments such as Paxlovid can provide substantial population-level impact by reducing hospitalizations. Given that these treatment regimes have to be administered early in the infection for it to reduce the severity of the disease, the analysis presented has of important value.

We thank the reviewer for this positive evaluation of our work.

My comments on the paper are as follows:

1. Hellwell data used is based on RT-PCR testing. Please state it clearly in the manuscript. It was also not clear what turnaround time for results was considered for the analysis, as that will determine how quickly people begin treatment.

We have included a statement in the in the first paragraph of our supplementary appendix reinforcing that while the Hellewell analysis uses PCR test data, test positivity estimates are inferred upon reflecting the diagnostic traits of LFT-type tests (i.e. assuming a lower CT threshold), which form the basis of our analysis.

2. The best strategy identified by the authors is testing every other day. The proportion of infected people who use drugs during this best strategy is the highest. While at the individual level, it makes sense. At the population level it may not be feasible. Therefore, in terms of policy, it is important to discuss who is being tested. So, an analysis that actually accounts for the proportion of the population who may get tested (this could be the population that is more likely to get hospitalized for instance), will provide a better insight into the population-level utility of the treatment strategies.

We now clarify in our abstract and the second and final paragraph of our main text that the focus of this analysis is indeed on a specific subset of the population who would most benefit from treatment, i.e. is at the greatest risk of facing severe disease. Therefore, the policy consistent with that which we assess here is in fact a focused, rather than indiscriminate, testing program. To address the effects of imperfect testing within this sub-population, we include an additional sensitivity analysis for which we simply weight estimated proportions benefiting from treatment by varying proportions of the population who tests. We find that, as with the other access-related parameters, estimated benefits depend critically on the proportion offered treatment, particularly for the more-frequent testing strategies (see paragraph 8). We additionally comment on the trade-offs between the regularity and availability of tests, such that scenarios defined by high test coverage but limited frequency are generally outperformed by scenarios defined by low test coverage but increased frequency (see paragraph 8).

3. In terms of feasibility, I am unsure why the author did not also evaluate the utilities of rapid antigen tests. I understand that Hellwell data was based on RT-PCR testing, however diagnostic sensitivity of rapid antigen tests relative to RT-PCR tests can be used to create a mapping to get relevant data for rapid antigen tests. The addition of such analysis will make findings from this study more valuable.

As the reviewer remarks, evaluating antigen tests is critical for any assessment of frequent testing strategies. As discussed above, this is indeed the focus of our analysis, given that we used test positivity estimates from the Hellewell analysis in a scenario that approximates the test detection abilities of antigen tests.

4. Were there any adjustments done to the Hellwell data to account for various or most current variants? Incubation periods, as well as the severity of the disease, vary by variant. Either accounting for them explicitly using estimates from literature or discussing the implications of not including them in the analysis is important.

We now include a scenario reflecting incubation periods that would better represent Omicron infections, i.e. characterized by a shorter incubation period distribution. While we find that our estimated weight risk ratios and proportions benefiting from treatment are higher and lower, respectively, than in our main scenario (as expected, given that a shorter time to symptom onset reduces the opportunity to test positive during a window when treatment is more effective), any differences between the two scenarios are minimal (see paragraph 6 of our main text). We also discuss in paragraph 10 that any differences in severity associated with a new variant would be unlikely to change estimated hospitalization risk ratios, as they are likely to lead to worse outcomes in both treatment and placebo groups. However, we also note that if under the new variant, treatment is effective for a longer period of time, then we would expect to see fewer gains associated with the more regular testing strategies. If treatment were to be effective under the same time window of efficacy, but to a greater extent, we would anticipate increased benefits for all strategies.

REVIEWER COMMENTS

Reviewer #1 (Remarks to the Author):

Equation 1. The summation at the top (on the second line) should be index-1 and not index. For example, if index =2 then for $i=0$ testing occurs 0, 2,4,6, ... days of infection. For $i=1$, then testing happens 1,3,5,7,... days of infection. When $i=2$, the initial sequence ($i=0$) is repeated but missing day zero of infection. Similarly, for Equation 2, the mean should be computed from 0 to index-1 when determining the proportion offered treatment. From my understanding, this notation could be updated with the revised notation referencing each series of testing times in general terms of a possible testing sequence.

Equation 3. The closing curly bracket ('}') is missing.
In the supplement and main text, there is still a reference to

Supplementary Figure S4 after the figure was removed from the manuscript.

Reviewer #2 (Remarks to the Author):

The presentation of the analysis by Menkir and Donnelly has improved since the first revision and (as mentioned previously) the work addresses an important question, that remains relevant. As this is a second revision, I limit my comments to concerns that relate to the validity of the methods. The text content of the main manuscript now reads very well and is at publication standard.

However, while the outcomes seem reasonable, I remain concerned about the equations, which are of course fundamental to the methods.

The equations remain difficult to understand and are incompletely described. In Equation 1, t_2 is not defined and the indexing for the product over testing negative throughout the testing period seem incorrect. I am unable to understand the rationale for then taking the product from i to 30-index in increments of index and then summing from $i=0$ to index. While, this may be a deficiency on my part, I did previously suggest this equation be more carefully described in the text because of the difficulty in understanding it, and I also note the Reviewer 1's concerns with the equations. Further, it seems impossible to take a product over the series $i, i+index, \dots, 30-index$, because this series will not reach 30-index in all cases. Similar considerations relate to equation 2. The code itself on GitHub is quite verbose and does not make optimal use of loops, such that even though this supports reproducibility, it does not help substantially in confirming the methods are correct. In Equation 3, more brackets are opened than closed.

Minor comment:

Supplemental Figure 1

Panels letters are not labelled on the figure panels (only in caption).

Reviewer #3 (Remarks to the Author):

Authors have addressed my comments appropriately. I congratulate them on this important work.

Response to referees

Reviewer #1:

Equation 1. The summation at the top (on the second line) should be index-1 and not index. For example, if index =2 then for i=0 testing occurs 0, 2,4,6, ... days of infection. For i=1, then testing happens 1,3,5,7,... days of infection. When i=2, the initial sequence (i=0) is repeated but missing day zero of infection. Similarly, for Equation 2, the mean should be computed from 0 to index-1 when determining the proportion offered treatment. From my understanding, this notation could be updated with the revised notation referencing each series of testing times in general terms of a possible testing sequence.

We thank the reviewer for pointing this out and suggesting abstracting how we enumerate the always test negative sequences. Following this suggestion, in equation 1 we use more generalized notation to refer to summing over possible always-test-negative sequences (highlighted in green), following the format we used to capture all sequences which include a final positive test, i.e. the first piece of equation 1:

$$\text{Weighted } RR_{\text{strategy } ts} = \frac{1}{\text{index}} \left\{ \sum_{seq \in tseq} \left[\prod_t^{t_{max}-1} Pr(\text{test } -)_t * Pr(\text{test } +)_t * RR_{t_{max}+delay-incubation.period} * P_{tmnt} \right] + \sum_{seq2 \in tseq2} \left[\prod_{t2}^{t2_{max}} Pr(\text{test } -)_{t2} * 1 \right] \right\}$$

[Equation 1]

where

$seq \in tseq$ denotes a given testing sequence seq - each defined by when testing is initiated - in the full set of possible sequences consistent with strategy ts ,

t is initialized as the first time point for each testing sequence,

t_{max} is the final day of that testing sequence,

$RR_{t_{max}+delay-incubation.period}$ denotes the risk ratio associated with the final day of that testing

sequence, back-shifted for the sampled incubation period, adding any test-to-treatment delays

P_{tmnt} denotes the the treatment proportion,

$seq2 \in tseq2$ denotes a given testing sequence in the set of always-test-negative sequences consistent with strategy ts

$t2$ is initialized as the first time point for each always-test-negative testing sequence, and

$t2_{max}$ is the final time point of that testing sequence

index = 2, 3, 7, and 14 for every other day, every three days, once a week, and once every two weeks, respectively

For Equation 2, we correct the upper bound of our range to index - 1.

Equation 3. The closing curly bracket (}') is missing.

We have removed the curly bracket, and now provide inner curved brackets, intermediate square brackets, and outer curly brackets for ease of interpretation.

In the supplement and main text, there is still a reference to Supplementary Figure S4 after the figure was removed from the manuscript.

We no longer reference Supplementary Figure S4 in either the main text or the supplement.

Reviewer #2:

The presentation of the analysis by Menkir and Donnelly has improved since the first revision and (as mentioned previously) the work addresses an important question, that remains relevant. As this is a second revision, I limit my comments to concerns that relate to the validity of the methods. The text content of the main manuscript now reads very well and is at publication standard.

We thank the reviewer for their acknowledgment of significant changes made in our revision.

However, while the outcomes seem reasonable, I remain concerned about the equations, which are of course fundamental to the methods.

The equations remain difficult to understand and are incompletely described. In Equation 1, t_2 is not defined and the indexing for the product over testing negative throughout the testing period seem incorrect. I am unable to understand the rationale for then taking the product from i to 30-index in increments of index and then summing from $i=0$ to index. While, this may be a deficiency on my part, I did previously suggest this equation be more carefully described in the text because of the difficulty in understanding it, and I also note the Reviewer 1's concerns with the equations. Further, it seems impossible to take a product over the series $i, i+index, \dots, 30-index$, because this series will not reach 30-index in all cases. Similar considerations relate to equation 2. The code itself on GitHub is quite verbose and does not make optimal use of loops, such that even though this supports reproducibility, it does not help substantially in confirming the methods are correct.

We appreciate the reviewer's concerns here and provide further clarification to alleviate any issues with interpretability. Specifically, we now define t_2 and replace our previous sum of products with more generalized notation referring to summing over the probabilities of all always-test-negative sequences (highlighted in green in equation 1 below):

*Weighted RR*_{strategy ts} =

$$\frac{1}{index} \left\{ \sum_{seq \in tseq} \left[\prod_t^{t_{max}-1} Pr(test -)_t * Pr(test +)_t * RR_{t_{max}+delay-incubation.period} * P_{tmnt} \right] + \sum_{seq2 \in tseq2} \left[\prod_{t2}^{t2_{max}} Pr(test -)_{t2} * 1 \right] \right\}$$

In Equation 3, more brackets are opened than closed.

We have removed the open bracket and now provide inner curved brackets, intermediate square brackets, and outer curly brackets for ease of interpretation.

Minor comment:

Supplemental Figure 1

Panels letters are not labelled on the figure panels (only in caption).

We now label the panels in Figure S1 in accordance with the legend.

Reviewer #3 (Remarks to the Author):

Authors have addressed my comments appropriately. I congratulate them on this important work.

We appreciate the reviewer's positive assessment of our work.

REVIEWERS' COMMENTS

Reviewer #1 (Remarks to the Author):

Minor:

Equation [2]. The indexing sequence in the product contains $i, i+index, \dots, 30-index$. For consistency, the authors could update this using their new notation introduced during the last revision.

Also, I missed in the last review mentioning that instead of writing $mean[0, index-1]$ in Equation [2], the authors could simply write $(1/index)$. These suggestions are more so to improve clarity for readers.

Response to referees

Reviewer #1:

Minor:

Equation [2]. The indexing sequence in the product contains $i, i+index, \dots, 30-index$. For consistency, the authors could update this using their new notation introduced during the last revision.

Also, I missed in the last review mentioning that instead of writing mean $mean[0, index-1]$ in Equation [2], the authors could simply write $(1/index)$. These suggestions are more so to improve clarity for readers.

We thank the reviewer for this additional feedback related to how we express the indexing. We have updated Equation 2 consistent with your suggestions.